# Multifilm Mass Transfer and Reaction Rate Kinetics in a Newly Developed In Vitro Digestion System for Carbohydrate Digestion

**DOI:** 10.3390/foods14040580

**Published:** 2025-02-10

**Authors:** Yongmei Sun, Jingying Cheng, Shu Cheng, Timothy A. G. Langrish

**Affiliations:** Drying and Process Technology Group, School of Chemical and Biomolecular Engineering, Building J01, The University of Sydney, Camperdown, NSW 2006, Australia; ysun2550@uni.sydney.edu.au (Y.S.); jche0791@uni.sydney.edu.au (J.C.); sche2348@uni.sydney.edu.au (S.C.)

**Keywords:** in vitro digestion, mass transfer, oral digestion model, carbohydrate digestion, mass transfer resistance, intestine model

## Abstract

Multifilm mass transfer theory has been used in conjunction with developing a new in vitro starch digestion model and applied to assessing starch digestion kinetics. One significance of this research is that this in vitro model has similar dynamics, such as similar Reynolds numbers for both in vivo and in vitro systems. In the in vitro intestine model, when the flow rate changes from 5.9 × 10^−6^ m^3^ s^−1^ to 1.0 × 10^−5^ m^3^ s^−1^ inside the intestine wall (inside the sausage casing), the *Re* number changes from 362 to 615. An oral digestion model, a stomach model, and an intestine model have been built to quantitatively understand reaction rate kinetics and two-film (or multifilm) mass transfer for carbohydrate digestion. This in vitro digestion system represents the oral mastication process to reduce the length scale of the test food, amylase inhibition in the stomach, and glucose generation and transport through the intestine wall according to the various emptying rates from stomach. Another dimensionless group, the Damköhler number (*Da*), has been calculated based on glucose measurements from this in vitro model, which show similar glycemic responses of the hydrolysis for banana and carrot with in vivo results. Another significance of this research is to distinguish a low GI food from a high GI one in this in vitro system and the possibility to estimate the GI value based on the glucose measurements.

## 1. Introduction

A significant challenge for in vitro methods of GI (glycemic index) measurement has been stated by Brand-Miller and Holt (2004) [1] in that they “are unlikely to detect the effect of differences in gastric emptying, nor the profound effect of viscosity on rate of absorption of the products of digestion”. The rate of gastric emptying after food consumption affects the digestion rate of food components in the small intestine [1]. However, it is possible that in vitro methods may be able to simulate the effects of different residence times and different mixing behaviour in between an in vitro mouth and an in vitro small intestine, hence simulating the effects of “differences in gastric emptying” [1]. An engineering perspective, such as mass transfer theory, may provide a qualitative understanding of in vitro digestion process and be helpful to improve the in vitro methods for GI measurement.

Carbohydrate digestion starts in the mouth, where salivary enzymes, such as amylase and invertase, hydrolyse carbohydrates into monosaccharides, such as glucose and disaccharides, including maltose and other shorter chain entities [2,3,4,5]. At the same time, the particle size or the length scale of a food is reduced by chewing and mastication [6,7,8]. However, many recent in vitro digestion models often exclude the oral digestion process and focus more on stomach models [9]. Table 1 summarizes recent in vitro physical models for food digestion study, where carbohydrate oral digestion is often absent from these models.

Correlations between salivary glucose and blood glucose have been found in some clinical studies. Some results indicated a significant positive correlation between fasting blood and fasting salivary glucose levels and postprandial blood and postprandial salivary glucose levels [14,15]. These findings are helpful for developing a non-invasive method for blood glucose tests, which are associated with monitoring postprandial blood glucose and Glycemic Index (GI) measurements [16,17,18,19]. However, correlations between salivary glucose and blood glucose have not always been found [20]. It may be worthwhile to develop an oral digestion model to mimic the oral digestion process based on the food physical characters and its reaction with digestive enzymes [13]. For carbohydrate digestion and its related glycemic response, it may be helpful to quantitatively understand the process of glucose generation (in oral digestion process and in the upper intestine) and the transport process from the mouth to the blood stream. In vitro digestion models provide the flexibility (avoiding high cost, ethical clearances, and operational barriers with in vivo studies) to study the digestive process in any point of the gastrointestinal system, which involves several mass transfer steps, such as the dissolution of food components, the movement of food components into digestive juices, then movement through the walls of the gastrointestinal tract. When in vitro digestion models mimic an in vivo digestion system, the dynamics, such as the Reynolds numbers and/or the mass transfer and transport behaviour (if the mass transfer coefficients could be calculated or measured) for both systems should be similar [21,22].

Starch hydrolysis may be catalyzed by both acids and by enzymes [23,24,25], specifically amylase, but amylase is not normally present in foods. Acid hydrolysis for carbohydrate digestion in stomach is not the dominant process, compared with enzymatic hydrolysis in the mouth and in the intestine [23,26]. Glucose from carbohydrate oral digestion arrives in the stomach with other components in a food bolus, such as oligosaccharides and disaccharides. Glucose has a relatively small molecular weight; thus, it has a greater mass transfer rate due to its small particle size, and it goes through the stomach quickly [27,28]. Glucose is usually emptied from the stomach within the first half hour of food digestion, flowing with about 30% of the total food bolus to the intestine [29]. The rest of the food bolus continues to mix with gastric acid, achieving a low pH and gradually emptying from the stomach into the small intestine (Liu et al., 2023a, 2023b, Abell et al., 2008) [29,30,31]. In the small intestine, the starch hydrolysis continues with pancreatic enzymes.

As shown in Figure 1, the overall mass transfer resistance in oral digestion includes two resistances, an internal mass transfer resistance inside a food and an external mass transfer resistance outside the food, when applying two-film mass transfer theory to understand carbohydrate digestion. After chewing and mastication in oral digestion process, starch granules are released from the broken plant cell walls, and the internal mass transfer resistance decreases, while the external mass transfer resistance may remain the same if the stirrer speed is the same. The salivary amylase hydrolyses the starch granules and generates glucose, disaccharides, and oligosaccharides. This food bolus goes through the stomach, where acid hydrolysis occurs, and amylase is inactive. In the small intestine (Figure 2), glucose generated by other salivary amylase in the food bolus and pancreatic amylase moves through the internal mass transfer resistance inside the intestine and is transported to the blood stream.

The overall mass transfer resistance, 1/K, may be estimated using Equation (1) [32].(1)$$\frac{1}{K}=\frac{1}{k_{int}\;}+\frac{\;H}{k_{ext}}$$

Here, *k_int_* is the mass transfer coefficient inside a food, *k_ext_* is the mass transfer coefficient outside the food (inside the mouth for oral digestion studies, and outside the intestine wall for intestine digestion studies), and *H* is the partition or equilibrium coefficient between the two phases (food and salivary enzymes for oral digestion process, or intestine wall and digestive juice).

Multifilm (and/or two-film) mass transfer theory has been applied to food digestion studies successfully [21,33]. Though it was developed for steady-state mass transfer, film theory is also applicable for unsteady-state situations if the interface region is thin, and the interface achieves quasi-steady state quickly with the bulk concentrations [21,32]. According to Equation (1), external mass transfer resistances always occur due to velocity differences between boundary layers and may not be negligible [21,34].

Mass transfer and enzymatic reaction (starch hydrolysis) occur simultaneously in this in vitro system. A dimensionless group, the Damköhler number (*Da*), may be used to assess the relative importance of mass transfer and chemical reaction in determining the overall reaction rate and time [35,36]. The *Da* number is defined as the ratio of the time constant for overall mass transfer to the time constant for reaction [37].

A newly developed in vitro system, developed here, has simulated the chemical aspects (such as enzymatic hydrolysis) and physical aspects (such as fluid velocity and geometry of the system) of food digestion. This in vitro model has been built using composite resin molar teeth (upper and lower dentures) in the oral digestion model, a beaker and stirrer system as the stomach model, and the sausage casing in the intestine digestion system to provide the boundary layers for the mass transfer resistances in the in vivo digestion process. It may be helpful to understand glucose transport and absorption by assessing the mass transfer resistances associated with the food digestion process.

The aim of this work is to quantitatively evaluate carbohydrate digestion in a novel in vitro model. The mass transfer coefficients (and resistances) have been analyzed in the new in vitro model and compared with in vivo conditions in the literature. The hypothesis is that the mass transfer coefficients are similar for both in vivo and in vitro conditions. An oral model and an intestine model have been built to mimic the digestion conditions by using composite resin molar teeth and sausage casings, which are the intestine walls from animals (hogs in this research). A stomach model has been set up in a beaker and stirrer system, using the plastic bag from the oral model. This stomach model connects the oral model and the intestine model to provide an in vitro digestion system by varying the velocities of food bolus in the gastrointestinal digestion processes. Glucose contents have been monitored during the carbohydrate hydrolysis in this in vitro system, focusing on the glycemic response of carbohydrate digestion. The dimensional analysis, such as estimates of the Reynolds numbers (Section 3.1, Section 3.2 and Section 3.4), shows that glucose mass transfer and transport behaviour are similar to the in vivo digestion processes. The interaction between mass transfer and reaction rate kinetics for carbohydrate hydrolysis in the small intestine has been assessed by the calculation of *Da* numbers based on the glucose content data (Section 3.4).

## 2. Materials and Methods

### 2.1. Materials

Glucose (soluble, Sigma-Aldrich, Bayswater, VIC, Australia), sucrose (Thermo Fisher Scientific, North Ryde, NSW, Australia), maltodextrin (dextrose equivalent 16.5–19.5, Sigma-Aldrich, Bayswater, VIC, Australia), invertase from baker’s yeast (S. cerevisiae) (Sigma-Aldrich, Bayswater, VIC, Australia), amylase from Bacillus licheniformis (lyophilized powder, 500–1500 units/mg protein, Sigma-Aldrich, Bayswater, VIC, Australia), SGF (Simulated Gastric Fluid, pH 1.7), and phosphate buffer (pH 6.9, 0.1 M) have been used in this research.

The test foods, bananas and carrots, have been purchased in a local market (Woolworth, St. Ives, NSW, Australia). The banana ripeness was about a scale of 6 (out of 7) according to Figure 1 in the literature [38], which puts it in the over-ripe range, highly ripe but not extremely ripe. The sausage casing has been purchased from KAA (Kay Apparel Australia, Kingaroy, QLD, Australia).

### 2.2. In Vitro Digestion Model

This in vitro digestion model includes a (molar) tooth model, a plastic bag (mimicking the oral cavity and the stomach), and an intestine model (using a sausage casing). It provides chemical and physical conditions, as well as physical aspects, for glucose generation in the mouth and in the small intestine and glucose transport from the mouth, through the stomach and the small intestine, to the blood stream. The test food was warmed up to a temperature of 37 °C and was mixed with digestive enzyme(s) in a plastic bag before the bag was hung between the molar teeth (the upper and lower dentures). The tooth model was operated for two minutes (more or less depending on the test foods) to mimic the mastication process in the oral digestion process. The bag containing the food bolus was then placed in a beaker and stirrer system at a temperature of 37 °C and a stirrer speed of 50 rpm (or 250 rpm), and a buffer solution was added into the bag. The glucose content was measured with a glucose meter (Accu-Chek Performa, Roche, Millers Point, Australia). After the glucose content measurement, SGF was added into the same bag to mimic the digestion conditions in the stomach, where amylase becomes inactivated.

The food bolus moves to small intestine at a certain gastric emptying rate [29]. In the intestine model, test foods (glucose, sucrose, maltodextrin, or starch) and enzyme(s) were placed inside the sausage casing; glucose contents from inside and outside the casing were monitored at certain time intervals. The detailed design of this in vitro model is described in the following section.

#### 2.2.1. Oral Digestion Model

This oral digestion model consists of a computer control system, gear pump, actuator, and composite resin molar teeth (upper and lower dentures) (Efucera-A, Tokyo, Japan). As shown in Figure 3, the computer system drove a gear pump, so the water moved between the actuators, and the hydraulic force in an actuator moved the molar teeth back and forth to mimic the chewing stresses. A separate bag was hung between the teeth, mimicking the oral cavity with each food sample and the relevant enzymes (e.g., carrot cubes and amylase). This tooth model was operated for about two minutes in each experiment to mimic the mastication and oral digestion process [39]. The dimensional analysis for comparing in vitro model and in vivo digestion system is reported in the Results and Discussion Section 3.1.

#### 2.2.2. Stomach Digestion Process

The same plastic bag with food bolus from the oral model (the oral digestion process) was placed in a beaker and stirrer system (Figure 4). The stirrer (at a speed of 50 rpm or 250 rpm) provided a peristaltic movement to the food bolus bag, mimicking the mobility of food in the stomach. Simulated gastric fluid with a pH of 1.7 was added into the bag to mimic the stomach digestion process. The Reynolds number was estimated for both the in vitro model and the in vivo digestion system in the Section 3.

#### 2.2.3. Intestine Digestion Model

Following oral digestion and mixing and retention in the acidic stomach environment, enzymatic carbohydrate digestion occurs in the small intestine. A sausage casing was used to mimic the glucose (with other foods and digestion juice) transport in the upper gastric digestion process. As indicated in Figure 5, gear pumps were used to transport the fluid inside and outside the sausage casing, which is suggested as an example of a real gut system. The sausage casing provides the mass transfer resistance for glucose transport through the intestine wall, and the intestine digestion model simulates the residence time for the carbohydrate digestion reaction in the small intestine. The dimensional analysis, such as the estimation of the Reynolds number and Damköhler numbers, was performed in the Section 3 for comparing the in vitro model and the in vivo digestion system.

### 2.3. Methods

Before placing the test sample and the amylase stock solution (in a plastic bag) into the tooth model, the bag of test food was warmed up at a temperature of 37 °C in a water bath to mimic food digestion conditions. This bag of test food and the amylase was then hung between the molar teeth for two minutes of mastication. A ratio of 2.4% *w*/*w* amylase to starch [40] has been used for this research. After the tooth model, a buffer solution (pH 6.9, 5 mL) was added into the bag to mimic the fluid, such as drinking water during an in vivo GI test. The glucose contents were measured using a glucose meter every five minutes for about twenty minutes, mimicking the food bolus transport through the mouth and esophagus.

About 5 mL SGF solution (pH of 1.7) was added into the same bag at a temperature of 37 °C and a stirrer speed of 50 rpm (or 250 rpm) after the tooth model to mimic the stomach process. The glucose contents were measured every ten or twenty minutes to understand the effects of the stomach digestion processes, such as amylase inhibition and starch hydrolysis rate reduction.

In the intestine model, carbohydrates with different degrees of polymerization, such as glucose, sucrose, and maltodextrin, were tested separately according to the various rates of stomach emptying. The test food, amylase, and invertase were placed inside the sausage casing, while glucose contents were measured outside the casing. The casing morphology was observed using SEM (scanning electron microscopy). The casing samples were coated by a Quorum-SC7620 Mini Sputter Coater (Quorum Technologies, Lewes, UK). Images of the coated samples were obtained via a Phenom-Prox SEM (Phenom-World, North Brabant, The Netherlands).

### 2.4. Data Analysis

For each step of the digestion system (both in vitro model and in vivo system), the Reynolds numbers were estimated by the following equation:(2)$$Re=\frac{\rho \;u\;d}{\mu }$$
where *ρ* is the density, *μ* is the viscosity, *d* is the equivalent diameter, and *u* is the velocity.

The flow rates or the velocity in this model have been controlled either by a gear pump or by a stirrer speed to mimic the in vivo flow rates during food digestion [24].

For *n*th-order reaction kinetics, the *Da* is defined as follows:(3)$$Da=\frac{time\;constant\;for\;overall\;mass\;transfer}{time\;constant\;for\;reaction}=K_{r}C_{0}^{n-1}\tau$$

Here, *K_r_* is the reaction rate constant (s^−1^), C_0_ is the initial concentration, *n* is the reaction order, and *τ* is the mean residence time (s), if the mean residence time is characteristic of the mass transfer rate in the solution. The exact formula for the *Da* may vary due to the various reaction rate law equations that apply to different reactions.

The glucose contents were measured by a glucose meter at 5–30 minutes’ time intervals. Each data point had three replicates, and standard deviations were calculated and reported, such as glycemic response curves in Section 3. Each experiment had at least two replicate trials. ANOVA (analysis of variance) was used in the statistical analysis of the test results to estimate the standard errors and to calculate the *p*-values in the GI data estimation for the test foods.

## 3. Results and Discussion

### 3.1. A Comparison of in Vivo Digestion System and in Vitro Models: Targeting Glucose Generation and Transport

Table 2 summarises the physical aspects (such as digestion timeline and geometry of each digestion region) and chemical aspects (such as enzymes and digestive juice for glucose generation) of carbohydrate digestion, which focuses on glucose generation (in the mouth and small intestine) and transport through the digestive system to the blood stream [22,41]. For the oral digestion process during in vivo GI measurements (ISO 26642:2010), 50 g of test food and 250 mL (up to 500 mL) of water are usually consumed within 15 min [42], and the velocity of the pharyngeal peristalsis (for a food bolus travel through the pharynx) is up to 0.4 m s^−1^ [43]. Given these input values, the Reynolds (*Re*) number may be estimated by Equation (2) as mentioned in Section 2.3.

Here *ρ* is the density of water, 993.37 kg m^−3^ at 37 °C; *μ* is the viscosity of water, 6.9 × 10^−4^ kg m^−1^ s^−1^ at 37 °C; *d* is the equivalent diameter of the mouth, 0.1 m in Table 2; and *u* is the velocity of the pharyngeal peristalsis, 0.4 m s^−1^ as defined above.

The *Re* for food bolus in the mouth is calculated to be 993.37 × 0.4 × 0.1/6.9 × 10^−4^ = 57,587, which is turbulent flow and is consistent with the result from [44]. According to the same equation, the *Re* numbers for the esophagus and the duodenum are about 1000, which are laminar flows [21]. The *Re* number is defined as the ratio of inertial forces to viscous forces within a fluid and is mainly a physical aspect that affects the mass transfer coefficient [45].

The in vitro digestion models should represent the physical aspects and chemical aspects of the in vivo digestion system from mouth to small intestine regarding glucose generation and transport. One hypothesis is that the mass transfer coefficients and/or reaction rate kinetics for both in vivo system and in vitro models are similar. The dimensional analysis continues in the following sections for each step of the carbohydrate digestion process.

Table 3 summarizes in vitro digestion models for food digestion processes in starch hydrolysis studies. The physical oral models include some fundamental parameters, such as saliva flow rate and mouth volume. These geometry and operating data are helpful to calculate relevant dimensionless groups, such as *Re*. Here the particle *Re* number is estimated because it is relevant to the transport phenomena of an object in a fluid, nutrients or enzymes to and from food particles. For instance, the oral model from Woda et al. (2010) [46] provides a mastication surface area, *A* = 170 mm^2^ = 1.7 × 10^−4^ m^2^; saliva flow rate, *Q* = 5 mL/min = 8.3 × 10^−8^ m^3^ s^−1^; and, thus, the velocity, *u* = *Q*/*A* = 8.3 × 10^−8^/1.7 × 10^−4^ = 4.9 × 10^−4^ m s^−1^. The equivalent length scale for the food (equivalent) particle is $\sqrt{170\;{mm}^{2}/\pi }$ = 0.007 m. According to Equation (2), the *Re* is estimated as 993.37 × 4.9 × 10^−4^ × 0.007/6.9 × 10^−4^ = 4.9, which is a flow regime in the intermediate region between Stokes Law (*Re* < 1) and the high Reynolds number region (*Re* > 500) [32,47]. This oral model from Woda et al. (2010) [46] focuses on the food particle size reduction in the oral digestion process, but there was no estimation of dynamic similarity, such as the *Re* number calculation, compared with the in vivo oral digestion system. Most of the published works on the physical intestine models have limited geometry data and/or operating conditions (Table 3). DGM (dynamic gastric model) results of starch hydrolysis study show a comparison of in vivo and in vitro data for the glycemic response without further explanation in terms of dynamic similarity, reaction kinetics, or mass transfer.

### 3.2. Mass Transfer and Reaction in the Lab-Built Oral Model

A performance qualification of the lab-built oral model has been tested using bananas and carrots, for which the in vivo GI is 51 ± 3 and 39 ± 4, respectively [54]. For bananas, the GI value varies at different stages of fruit ripeness. In general, overripe bananas have a medium GI value, such as 57 [55]. If the ripeness is extremely high, the GI value may be in the high GI range (greater than 69) because the low DP (degree of polymerization, 1–2) carbohydrates increase and the blood glucose levels are high during digestion process [56,57]. As shown in Figure 6, test foods and amylase went through a mastication process, they were then transferred to the beaker and stirrer system, and the buffer solution was added before measuring glucose contents. During mastication, the plant cell walls of the test food were broken, and the internal mass transfer resistance was reduced. After mastication, the internal mass transfer resistance decreased significantly in the beaker and stirrer system, and the overall mass transfer resistance was then mainly contributed by the external mass transfer resistance, 1/*K_overall_* ≈ 1/*K_exteranl_*. In the beaker and stirrer system, the tube *Re* number is estimated by Equation (4) [58]:(4)$$Re=\frac{\rho ND^{2}}{\mu }$$

Here *Re* is the impeller Reynolds number, *N* is the stirrer speed, *D* is the impeller diameter, *ρ* is the fluid density, and μ is the fluid viscosity (*μ* and *ρ* have been defined above, *μ* = 6.9 × 10^−4^ kg m^−1^ s^−1^; *ρ* = 993.37 kg m^−3^). *Re* = 993.37 × (50/60) × 0.04^2^/6.9 × 10^−4^ = 1919.6, which is more than 10 and less than 10,000, and the flow is transitional [58]. Due to the food bolus bag being far from the impeller (stirrer), most of the flow situation tends to be laminar, which is consistent with the in vivo flow patterns in the esophagus and the duodenum (as discussed in Section 3.1).

As shown in Figure 7, the actual glucose concentrations were collected from the oral digestion of test foods (banana and carrot) using the oral model (Figure 6) while the fitted concentrations were achieved by fitting the experimental concentrations to Equation (5), which is the expected response for a constant mass transfer coefficient [59]. The time constant has been calculated using the Least-Squared-Error (LSE) method.(5)$$C=C_{max}\left[{1-e}^{\left(-\frac{t}{\tau }\right)}\right]$$

Here *C_max_* is the equilibrium concentration of glucose, and *τ* is the time constant for starch hydrolysis.

The oral digestion process takes a short period of time, usually less than one minute. If it is assumed that the residence time in this oral model is half a minute, the frequency of mastication is somewhere between seven chews and fourteen chews within half a minute for the test foods (bananas and carrots). The initial size of the test foods is one cubic centimetre for both bananas and carrots. A high frequency of mastication results in greater glucose contents for banana hydrolysis (oral digestion process), but not for carrots (Figure 7).

Based on the glucose concentrations in Figure 7, the overall mass transfer coefficients for the oral digestion process of the rest foods have been estimated by Equation (6), and the results are summarized in Table 4. At the same mastication frequency, the overall mass transfer coefficients for carrot hydrolysis are significantly lower than those for banana hydrolysis. Considering that the external mass transfer coefficients are similar, the overall mass transfer coefficient is mainly affected by the internal mass transfer resistance according to Equation (1). This may be explained by the microstructure of the plant cell walls, since the banana cell wall is more permeable than others [60,61], while carrot has a relatively firm texture [62,63], thus carrot is likely to have a greater mass transfer resistance than banana for the same size of the test foods. It is also possible to explain the results in Figure 7 by the carbohydrate variety and contents in bananas and carrots. Carrot is a root vegetable with a relatively low starch content, compared with banana. The starch in carrot is not easily degraded into low DP carbohydrates, such as glucose and sucrose, while most of the carbohydrates in banana tend to convert into low DP carbohydrates during the fruit-ripening process [64,65,66]. The following equation was used to obtain the experimental value of the overall mass transfer coefficient, *K*, from the slope of the experimental glucose concentration–time curve (*dC*/*dt*).(6)$$K=\frac{V}{A}\;\frac{dC}{dt}\frac{1}{Csat}$$

The other variables in this equation are the saturation concentration (*Csat*) of the glucose in the solution, the volume of the solution (*V*), and the interfacial area between the test foods and the solvent (*A*).

In terms of two-film or multifilm mass transfer theory, Equation (1) in Section 1 was used to relate internal and external mass transfer resistances to the overall mass transfer coefficient. Here, 1/*k_int_* is the internal mass transfer resistance, *H* is the partition or equilibrium coefficient for the solute between the two phases, and 1/*k_ext_* is the external mass transfer resistance.

The Damköhler number (*Da*) is the ratio of the chemical reaction rate to the mass transfer) rate occurring in a system [67]. It may also be calculated as the ratio of a transport time to a reaction time [68]. If Da >> 1, the reaction rate is much greater than the mass transfer rate. By contrast, if Da << 1, mass transfer occurs much faster than the chemical reaction. Dimensionless groups, such as the Damköhler number (*Da*), express the ratios of different physical processes (here mass transfer and chemical reaction), and these groups are important in engineering for scale up.

In Table 5, the estimation of the *Da* numbers shows that the reaction rates for the test foods are greater than the mass transfer rates, so more chewing and mastication results in greater *Da* numbers. For the oral digestion process of bananas, the reaction rate is far greater than the mass transfer rate due to its greater cell wall permeability for the diffusion of the digestive enzymes [60]. Compared with bananas, carrots have lower reaction rates and higher mass transfer rates because of the firm cell walls. These in vitro results in the oral digestion process (before the digestion in the stomach) show similar trends to the in vivo GI values; for instance, the GI value for bananas (51 ± 3) is higher than that for carrots (39 ± 4).

### 3.3. Understanding Amylase Inhibition in the Stomach Model

After the food bolus arrives in the stomach and mixes with acidic gastric juice, the enzyme amylase becomes inactive, and starch hydrolysis is inhibited [41]. As shown in Figure 8, the test foods, banana and carrot (in the presence of amylase), showed this inhibition, and the glucose contents were almost constant after the SGF solution was added into the food bolus bag. Glucose and/or other small sugars (low degrees of polymerization), such as disaccharides and oligo saccharides from starch hydrolysis in the oral digestion process, have small molecular weights and are water soluble, so they are most likely to be found in the liquid layer (close to the outlet of the stomach) and emptied quickly from the stomach (Figure 1 from Liu et al., 2023a [30]). The rest of the food bolus remains in the stomach, mixing with digestive juices and reducing the particle sizes, and is emptied from the stomach gradually. This process is important for the GI measurement both in vitro and in vivo [36]. A realistic in vitro stomach model must represent the stomach emptying rate for the in vitro GI measurement.

In the following section, test carbohydrates have been selected according to the different degrees of polymerization in order to mimic the different stomach emptying rates. Monosaccharide (glucose), disaccharides (sucrose), and oligosaccharides (maltodextrin) have been tested in the lab-built intestine model to understand the glucose generation and transport in the small intestine.

### 3.4. Model Time Constants for Mass Transfer and Reaction in the Lab-Built Intestine Model

As shown by the typical results in Figure 9, the lab-built intestine model has been tested for a performance qualification using glucose. The glucose solution inside the intestine wall (the sausage casing) flows along inside the casing, and the glucose contents decrease over time, while glucose transfers from the inside casing to the outside casing due to the difference in the glucose contents inside the casing and outside the casing, resulting in the glucose contents outside the casing increasing over time.

At the same time, the sausage casing used in the intestine model represents the geometry of the real gut system. As shown in Figure 10, the SEM images of both dry and soaked casings indicate the villi on the intestine wall for the in vitro system, and the surface area for absorption also increases in the in vivo condition, such as the case for the soaked casing. For instance, according to clinical results, Table 1 from [22], the absorption area in the duodenum is 0.09 m^2^, which is two and half times larger than the apparent surface area (π *d L* = π × 0.05 × 0.25 = 0.04 m^2^) if the duodenum is assumed as a smooth cylinder. For this intestine model, the diameter of the sausage casing (*d*) is 30 mm, and the length of the casing in the pan (*L*) is 200 mm, so the interfacial area (*S*) is π *d L* = π × 30 × 200 × 10^−6^ = 0.02 m^2^ without considering the villi. If applying the real gut system geometry into this calculation, the absorption area for this intestine model is approximately 0.05 m^2^ (0.02 × 2.5).

As estimated in Section 3.1, the *Re* number in the duodenum is about 1000 during the in vivo digestion process, and the food bolus in this part of the GI system is in laminar flow. For this lab-built intestine model, the glucose flow inside the sausage casing had a *Re* number of 615 when the flow rate was 1.0 × 10^−5^ m^3^ s^−1^ (and 362 at a flow rate of 5.9 × 10^−6^ m^3^ s^−1^), which is laminar flow as well. The in vivo system and in vitro intestine model have similar fluid dynamic behaviour.

The time constant is about 1.23 h, according to the calculation from Langrish (2022) [21] based on Table 2. The current in vivo method to determine the glycemic response of a test food is to measure small numbers of blood samples from the finger over a period of two hours [69]. According to the digestion time (in vivo timeline in Table 2), the test food is quite likely to have passed through the stomach into the duodenum within a measurement time of two hours [22,29]. The dimensional analysis of food digestion process focuses on the in vivo fluid in the duodenum (compared with the in vitro intestine model). For this in vitro model, the glucose contents outside the casing are relevant to the in vivo GI method, which measures the glucose transport through the intestine wall (to the blood stream). For the intestine digestion process, the intestine wall (sausage casing in this in vitro model) significantly reduces mass transfer rates, because this casing wall adds another film to the whole digestion process, and it adds to the series of mass transfer resistances (Table 6). These findings are consistent with other results in food digestions studies, where applying mass transfer theory to interpret the results [21,35].

For sucrose and maltodextrin (dextrose equivalents 16.5–19.5), the mass transfer rates are significantly higher than the reaction rate during the intestine digestion process. As shown in Table 7, *Da* numbers are much less than unity (1), so the mass transfer process of the digestive enzymes is the rate-limiting step for carbohydrate hydrolysis in the intestine digestion process.

For carbohydrate digestion, both oral and intestine hydrolysis processes are important for the glycemic response during GI measurement. Figure 11 shows the results of glucose contents outside the sausage casing for single food (bananas and carrots) digestion in the lab-built in vitro system (oral model and intestine model), and the trends and orders of magnitudes are consistent between the results in the oral model (Figure 7) and the in vivo data (the GI value for banana is higher than that for carrot) [54,61]. For the test food, banana, different degrees of ripeness may result in different carbohydrate components, such as more low-DP carbohydrates for ripe banana, thus giving higher GI values when banana ripens. If a banana is overripened, the GI value is likely to be in the medium GI range (56–69) [55]. Compared with banana, carrot has a relatively low GI value. The results in Figure 11 show that this in vitro system can distinguish a low GI carrot from a higher GI banana, as the in vivo GI method does [55].

GI is a percentage of the area under the glucose response curve for a test food when the same amount of glucose is taken as a reference [70]. Figure 12 and Figure 13 show the banana and carrot glycemic response curves with the same amount of carbohydrates as glucose. The GI values may be estimated by the area under the curve (AUC) method based on the glucose contents data over 2 h (or 120 min). For example, $\int_{0}^{120}f\left(x\right)dx$ is used for AUC calculation, $f\left(x\right)$ is the glucose concentration (*C*, mg/mL), and dx is the hydrolysis time (*t*, minute) in this study. For the glycemic response curves in Figure 12 and Figure 13, AUC is $\int_{0}^{120}C_{s}(1-exp\left(-t/\tau \right))dt$, where *C_s_* is the maximum glucose content estimated in the previous section, and *τ* is the time constant for the measurement. In this way, the estimated GI value for banana has been found to be 47 ± 3 (R > 0.9, *p* < 0.001) in Table 8, which is consistent with the in vivo data in the literature [54,55]. The GI value for carrot is estimated as 12 ± 2 (R > 0.9, *p* < 0.001) in Table 8, and low GI values for raw carrots (16,23) have been reported in some in vivo studies [55,71]. The variation of GI estimation may be due to the complexity of mimicking the chemical aspects of the digestion system, which is also challenging for the in vivo method.

## 4. Conclusions

This lab-built in vitro digestion system in this study, mimicking the in vivo digestion process, has two key aspects: the physical representation of digestion fluid dynamics, and the chemical representation of digestive enzymes and carbohydrate hydrolysis. The Reynolds numbers in this in vitro system are similar to those in the in vivo digestion processes, such as oral digestion, stomach digestion, and (small) intestine digestion. This dynamic similarity indicates that an in vitro model represents the in vivo digestion process reasonably well in physical terms.

Applying multifilm mass transfer theory for carbohydrate hydrolysis study in this in vitro system allows the quantitative results to be assessed for the model qualification, such as performance qualification (glucose transport) and carbohydrate hydrolysis with different DP samples (sucrose, maltodextrin, single foods). The test foods (bananas and carrots) show similar trends in the glycemic response during the hydrolysis processes in this in vitro system.

This research has investigated the hydrolysis process for carbohydrate-rich foods. Though the estimation of GI values for bananas and carrots using AUC from hydrolysis is not perfect, the results are not far from literature values. Future work may focus on further improving the realism of the chemical conditions for the hydrolysis. It may also be worthwhile to consider a further investigation of mass transfer and reaction kinetics for the digestion of protein-rich foods.

## Figures and Tables

**Figure 1 foods-14-00580-f001:**
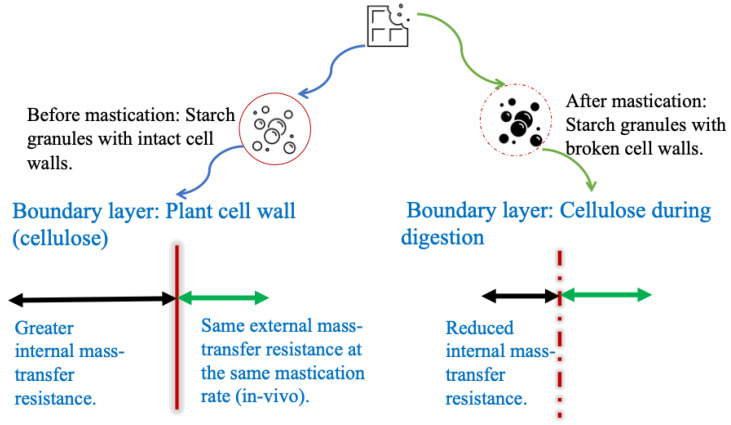
Diagram of two-film mass transfer in carbohydrate oral digestion. The internal mass transfer resistance decreases after mastication (shown in black arrow lines), while the external mass transfer resistance (shown in green arrow lines) keeps constant at the same mastication rate (in vivo).

**Figure 2 foods-14-00580-f002:**
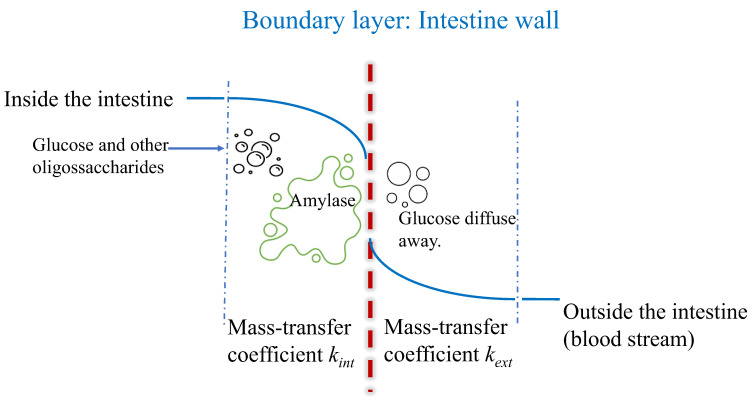
Diagram of two-film mass transfer in carbohydrate intestine digestion.

**Figure 3 foods-14-00580-f003:**
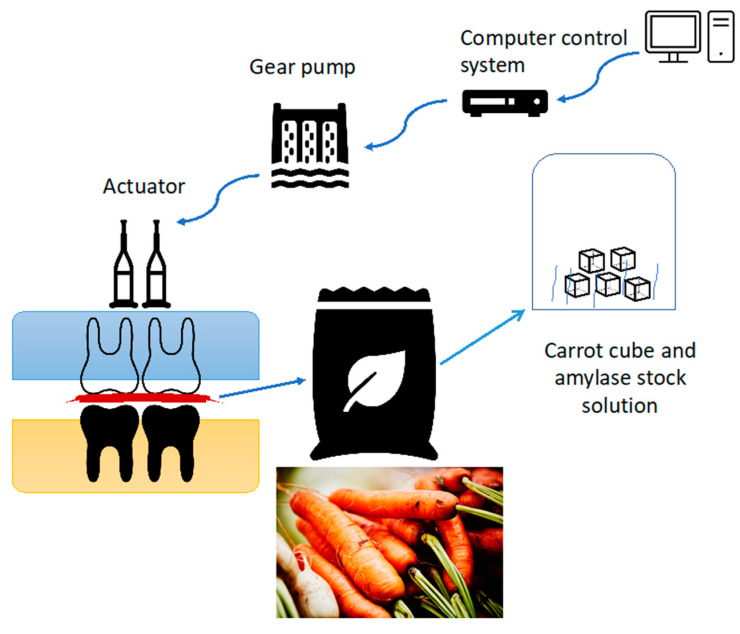
Diagram of the oral digestion model mimicking the mastication in the oral digestion process.

**Figure 4 foods-14-00580-f004:**
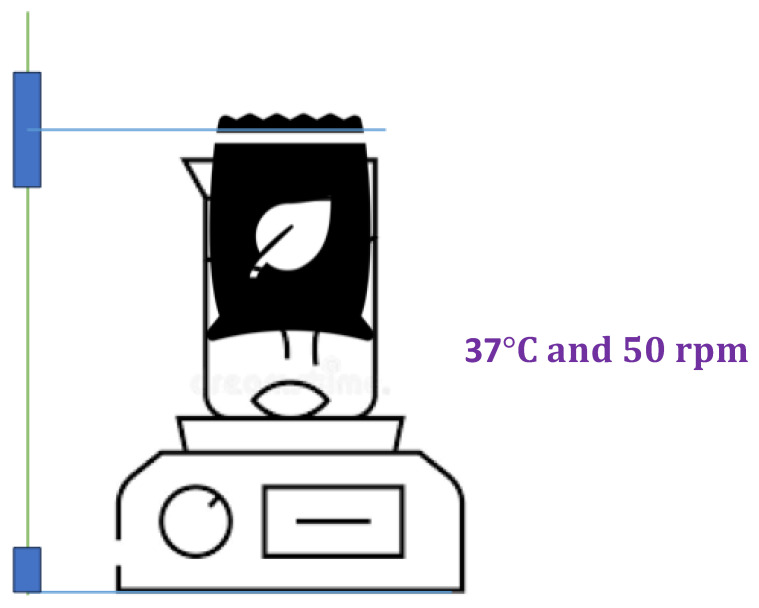
Diagram of the stomach digestion model mimicking the residence of food bolus in the stomach and carbohydrate digestion inhibition.

**Figure 5 foods-14-00580-f005:**
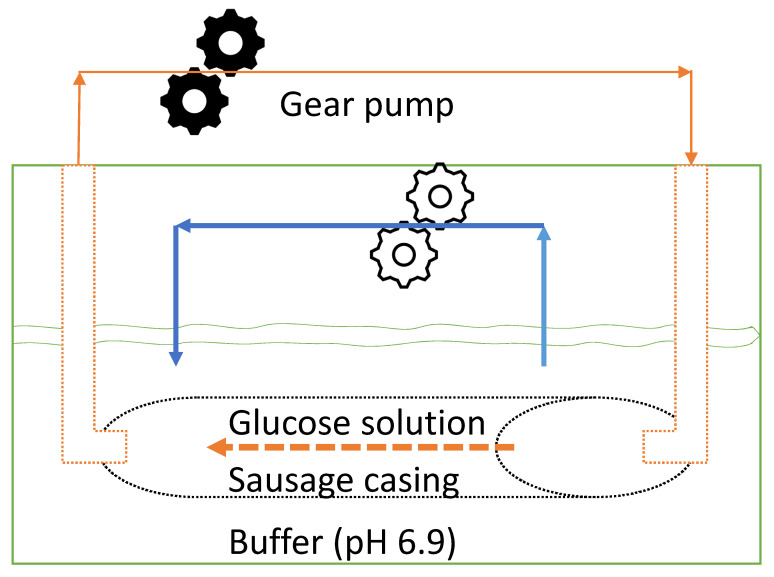
Schematic diagram of the intestine digestion model designed to replicate glucose generation and transport dynamics. The orange lines indicate the flow inside the intestine (the sausage casing), and the blue lines indicate the flow outside the intestine.

**Figure 6 foods-14-00580-f006:**
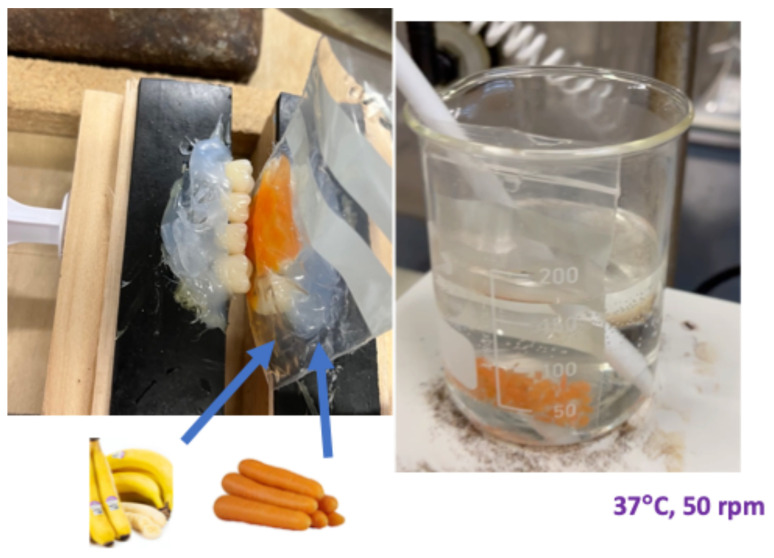
Images of the oral digestion model including a molar tooth model to mimic the mastication and food bolus formation and a beaker and stirrer system to mimic the transport phenomena after swallowing.

**Figure 7 foods-14-00580-f007:**
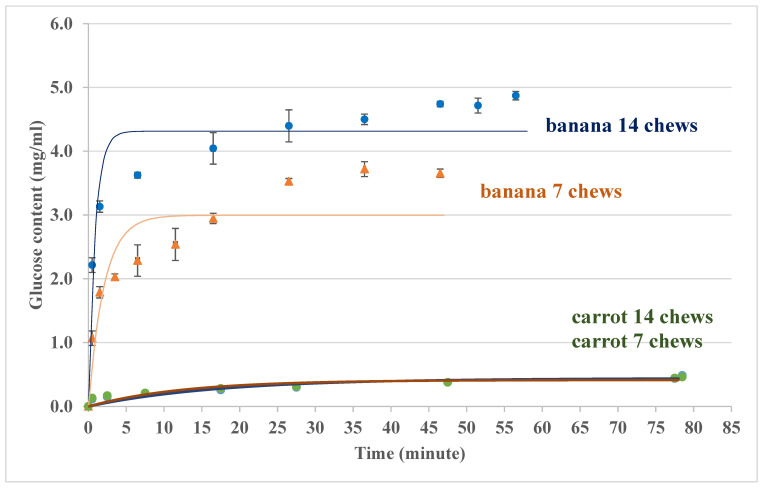
Glucose content data from banana and carrot oral processing using the oral model and the beaker and stirrer system at 50 rpm and a pH of 6.9.

**Figure 8 foods-14-00580-f008:**
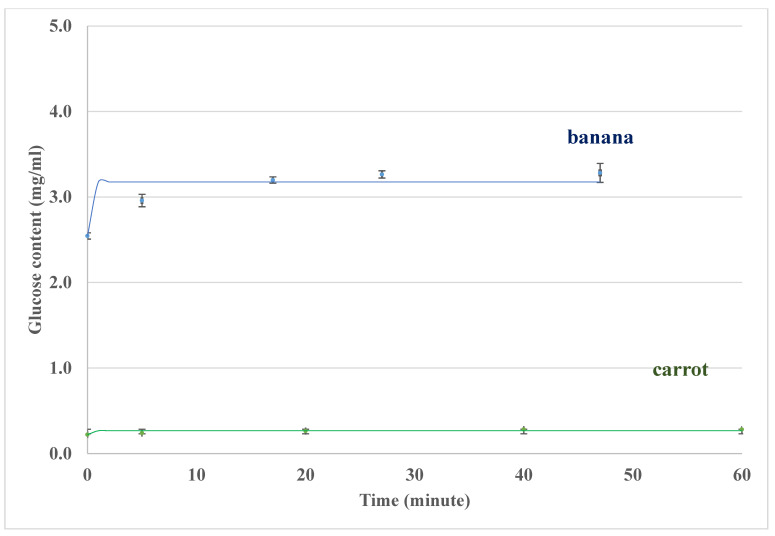
Glucose content data from banana and carrot stomach processes using the beaker and stirrer system at 50 rpm.

**Figure 9 foods-14-00580-f009:**
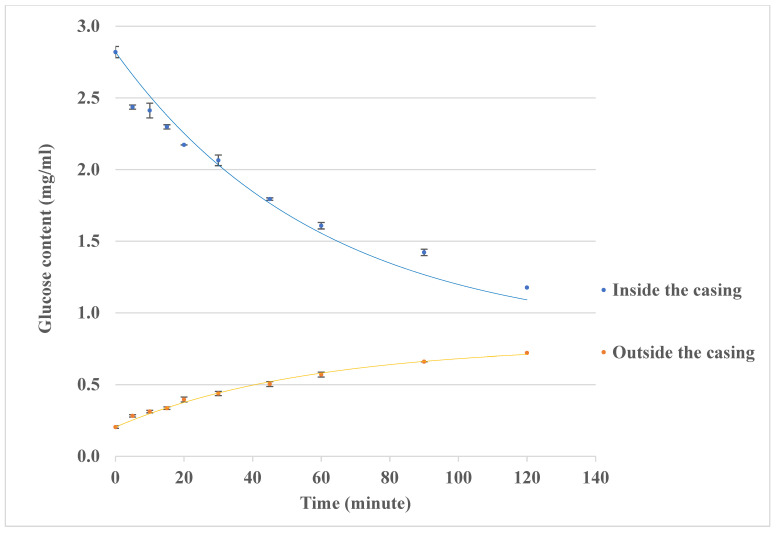
Performance qualification using glucose in the lab-built intestine model. The glucose transfers from inside the sausage casing to the outside of the casing, mimicking the in vivo glucose transport through the intestine wall to the blood steam. The glucose contents inside the casing decrease, and the glucose contents outside the casing increase.

**Figure 10 foods-14-00580-f010:**
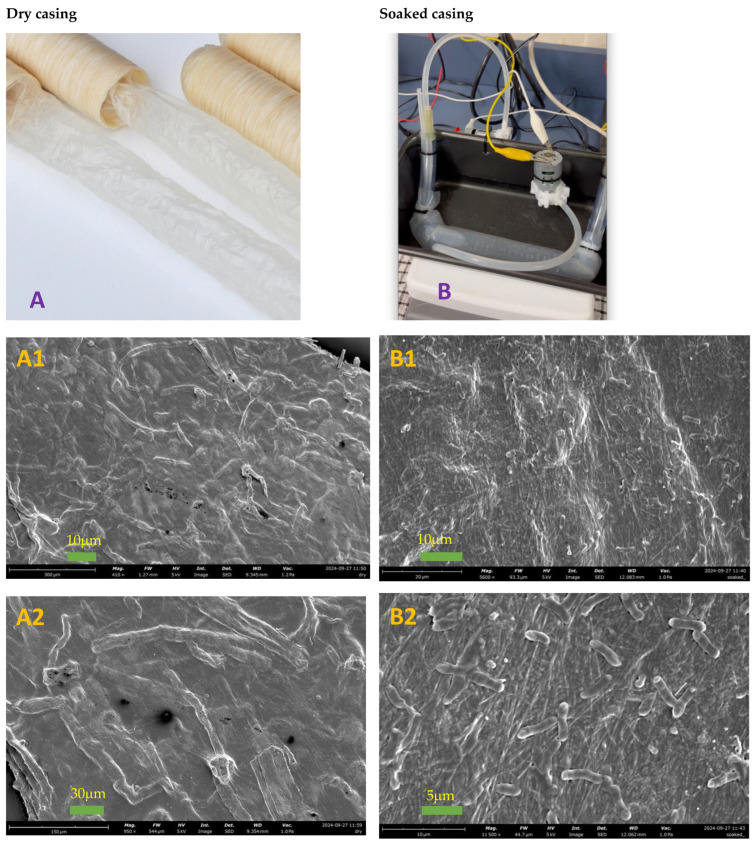
The lab-built intestine model (**B**) using the sausage casing (**A**) and the SEM images of the dry casing (**A1**,**A2**), and the soaked casing (**B1**,**B2**) showing the villi on the intestine wall, which increases the surface area after soaking.

**Figure 11 foods-14-00580-f011:**
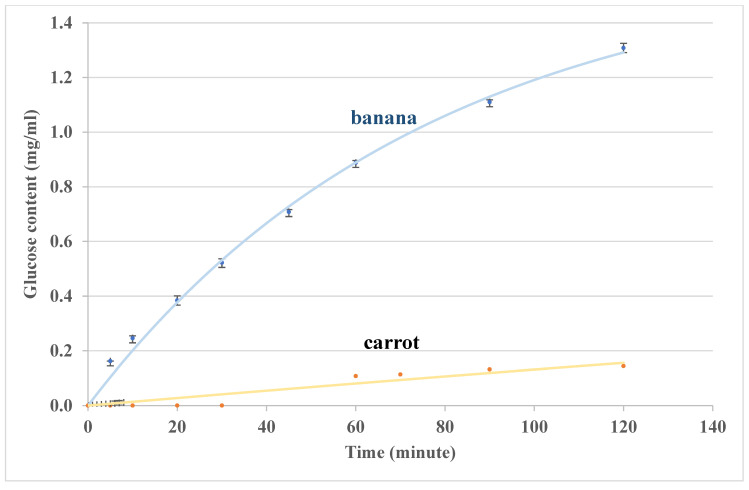
Glucose contents outside the sausage casing during single foods (bananas and carrots) digestion in the lab-built in vitro system (oral model and intestine model). Some error bars are smaller than the symbols on the figure.

**Figure 12 foods-14-00580-f012:**
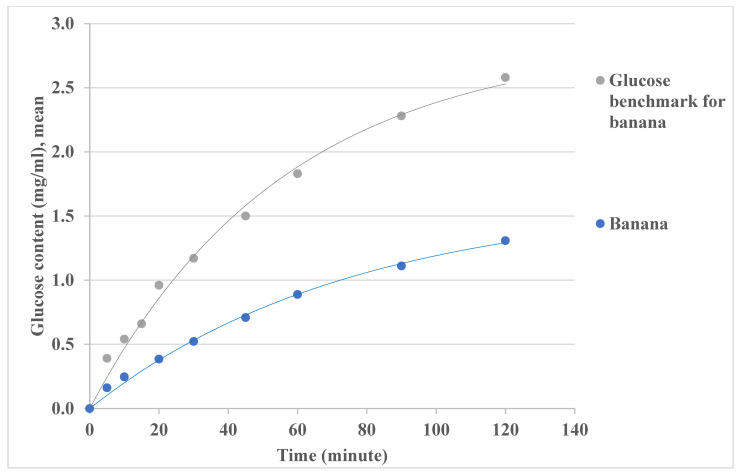
Glucose contents for banana GI estimation using the area under the curve (AUC).

**Figure 13 foods-14-00580-f013:**
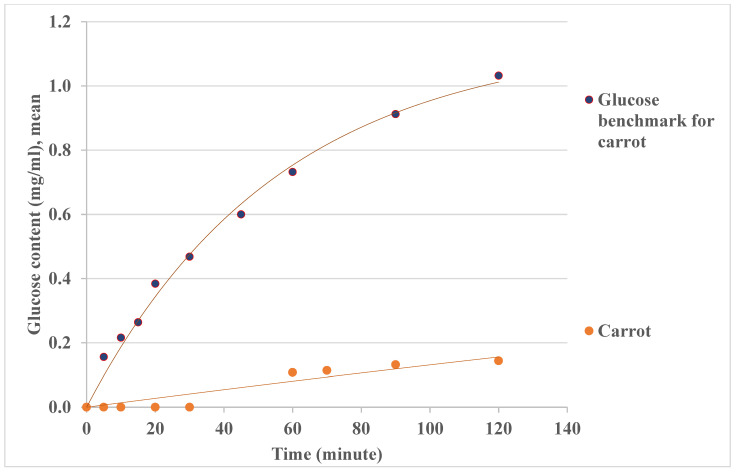
Glucose contents for carrot GI estimation using AUC.

**Table 1 foods-14-00580-t001:** Review of recent in vitro physical models and analysis of oral digestion processes within these models.

Physical Digestion Models	Oral Digestion Process	Study Focus	Reference
An electro-mechanical system for food oral processing research.	This chewing system mimics human chewing functions (mastication) based on electro-mechanical controls.	Food texture and flavour analysis using a model mouth (real shaped teeth and multi-point saliva injections).	[10]
Semi-dynamic in vitro digestion method suitable for food	Shaking incubator and mincer	This in vitro model includes a food oral digestion phase, especially in terms of chemical aspects, but there is less focus on the geometry or the physical aspects of the oral digestion process.	[11]
A digestion chip	Oral digestion process not discussed	This miniaturised digestion system gives good control of pH and temperature, during the two main phases of digestion (gastric and intestinal).	[12]
Dynamic in vitro models	Non model mimics the in vivo oral digestion process, such as chewing, mastication, food bolus formation, and the interaction of food components and salivary enzymes.	Nine dynamic models have been reviewed on the mechanism of (solid) food breakdown. There are no standardized methods to validate the models.	[13]

**Table 2 foods-14-00580-t002:** Physical and chemical aspects of glucose generation and transport in the digestive system.

Digestion Region	Physical Aspects				Chemical Aspects		
	In Vivo Timeline	A, Absorption Area (m^2^)	L, Length (cm)	D, Diameter (cm)	Enzyme	pH	Digestive Juice
Mouth	Under free will, or within 15 min for 50 g test food for in vivo GI measurement	~0.07	15–20	10	Amylase, invertase	5.8–7.1	Saliva
Esophagus	9–15 s	~0.02	25	2.5	--	5.6	--
Stomach	0.5–4.5 h	~0.11	20	15	--	1.0–3.5	Gastric fluid
Small intestine, including duodenum	1–4 h	~120 (~0.09 in duodenum)	635–655 (25 in duodenum)	5	Amylase, invertase	6.9–7.9	Pancreatic juice and intestinal fluid

**Table 3 foods-14-00580-t003:** Analysis of in vitro digestion models for starch hydrolysis studies.

In Vitro Models	Description	Geometry Data		Operating Conditions		References
Oral models	AM2 (physical model)	Masticatory surface area: 170 mm^2^	Mouth volume: ~15 cm^3^	Saliva flow: 0.5–5 mL/min	Not available	[46]
	A chewing simulator (physical model)	Forces: up to 250 N	Working volume: up to 50 mL	Saliva inlet: up to 5 mL/min	Biting speed: 75 mm/s	[48]
Intestine models	TIM-Carbo (physical model)	Not available	Not available	Not available	Not available	[49,50]
	DGM (physical model)	Capacity/Volume: 800 mL	Not available	Duodenal Processing in an orbital shaker: 170 rpm	Not available	[51,52]
	CSTR, PFR (Mathematical model)	Radius: 1.8 cm	Length: 2.85 m	Velocity: 1.7 × 10^−4^ m/s	Viscosity: 0.001–10 Pa·s	[53]

**Table 4 foods-14-00580-t004:** Estimates of mass transfer coefficients and mass transfer resistances for oral digestion process of the test foods (banana and carrot).

Samples	Slope of the Glucose Concentration–Time Curve (dC/dt) (mg/mL/s)	Estimated Maximum Glucose Concentration (mg/mL)	Estimated Overall Mass Transfer Coefficient (m/s)	Estimated Overall Mass Transfer Resistance (s/m)
Banana, 7 chews	(2.1 ± 0.1) × 10^−2^	3.1 ± 0.05	(9.3 ± 0.6) × 10^−4^	(1.1 ± 0.06) × 10^3^
Banana, 14 chews	(3.9 ± 0.1) × 10^−2^	4.3 ± 0.07	(1.2 ± 0.04) × 10^−3^	(8.5 ± 0.3) × 10^2^
Carrot, 7 chews	(1.1 ± 0.07) × 10^−3^	0.4 ± 0.005	(4.6 ± 0.3) × 10^−5^	(2.2 ± 0.1) × 10^4^
Carrot, 14 chews	(1.3 ± 0.07) × 10^−3^	0.4 ± 0.008	(5.2 ± 0.3) × 10^−5^	(1.9 ± 0.1) × 10^4^

**Table 5 foods-14-00580-t005:** Damköhler numbers estimation for oral digestion process of the test foods (bananas and carrots) by this in vitro system.

Samples	Time Constant for Overall Reaction (Minutes)	Time Constant for Overall Mass Transfer (Minutes)	Da (Time Constant for Overall Mass Transfer/Time Constant for Reaction)
Banana, 7 chews	2.2 ± 0.4	(2.0 ± 0.1) × 10^4^	(9.6 ± 0.6) × 10^3^
Banana, 14 chews	0.9 ± 0.05	(2.5 ± 0.08) × 10^4^	(2.8 ± 0.09) × 10^4^
Carrot, 7 chews	17.0 ± 1.9	128 ± 8	7.2 ± 0.4
Carrot, 14 chews	12.3 ± 1.2	144 ± 8	11.6 ± 0.6

**Table 6 foods-14-00580-t006:** Estimates of mass transfer coefficients and mass transfer resistances for intestine digestion processes with different DP carbohydrates (sucrose and maltodextrin), which represent different stomach emptying rates. The hydrolysis with amylase and invertase was conducted at the same test conditions.

Test Sample	Slope of the Glucose Concentration–Time Curve (dC/dt) (mg/mL/s)	Estimated Maximum Glucose Concentration (mg/mL)	Estimated Overall Mass Transfer Coefficient (m/s)	Estimated Overall Mass Transfer Resistance (s/m)
Sucrose	(4.0 ± 0.2) × 10^−4^	1.8 ± 0.05	(6.3 ± 0.09) × 10^−7^	(1.6 ± 0.02) × 10^6^
Maltodextrin	(1.2 ± 0.3) × 10^−4^	0.26 ± 0.05	(1.3 ± 0.3) × 10^−6^	(8.0 ± 2) × 10^5^

**Table 7 foods-14-00580-t007:** Damköhler numbers estimation for intestine digestion process of the test foods (sucrose and maltodextrin) by this in vitro system.

Samples	Time Constant for Overall Reaction (Minutes)	Time Constant for Overall Mass Transfer (Minutes)	Da (Time Constant for Overall Mass Transfer/Time Constant for Reaction)
Sucrose	56 ± 3	(1.5 ± 0.02) × 10^−3^	(2.7 ± 0.2) × 10^−5^
Maltodextrin	56 ± 9	(3.2 ± 0.8) × 10^−3^	(6.0 ± 2) × 10^−5^

**Table 8 foods-14-00580-t008:** GI value estimates for banana and carrot hydrolysis in the in vitro digestion system using the AUC method (95% confidence intervals).

Test Food	AUC Glycemic Response for Test Food (mg/mL·min)	AUC for Glucose as a Reference Food (mg/mL·min)	GI (%)
Banana	94.12–99.67	196.74–215.13	47 ± 3
Carrot	8.50–10.89	78.70–86.05	12 ± 2

## Data Availability

The original contributions presented in this study are included in the article. Further inquiries can be directed to the corresponding author.

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
