# Peer review of "Multifilm Mass Transfer and Reaction Rate Kinetics in a Newly Developed In Vitro Digestion System for Carbohydrate Digestion"

_foods, 2025, doi:10.3390/foods14040580_

Round 1

Reviewer 1 Report

Comments and Suggestions for Authors

The study of simulating the internal digestion of the body in this article is a very interesting research topic.

1.       Add the significance of this study to the abstract. There is always a certain difference between in vitro digestion and in vivo digestion, and it is recommended to elaborate on the guiding significance of in vitro digestion.

2.       The author can add the significance of simulating in vitro digestion in the forefront introduction.

3.       The author should briefly introduce the characteristics of the intestinal model at the forefront.

4.       The author explained that the results of in vivo and in vitro kinetic digestion are similar, and the author can provide the results of the correlation coefficient.

5.       Does the use of standardized banana samples or commercially available bananas by the author have a certain impact on the results of the repeated experiment.

6.       Is the blood glucose response in the conclusion consistent with the blood glucose response in the human body, or how accurate is its performance.

7.       The format of references needs to be standardized, and it is recommended to cite and compare them with digestive literature in vitro for analysis. DOI: 10.1016/j.fbio.2024.104499. 10.1016/j.lwt.2024.116869.

Reviewer 2 Report

Comments and Suggestions for Authors

The objective of this paper is to understand in-vitro carbohydrate digestion quantitatively in a newly developed in-vitro digestive system. The study aims to analyze mass transfer coefficients and resistances under both in-vivo and in-vitro conditions, testing the hypothesis that these coefficients are similar for both. The research incorporates an oral model, a stomach model, and an intestine model, using innovative designs to mimic physiological conditions and evaluate glucose generation and transport dynamics.

Authors are advised to make the following adjustments to the paper:

Abstract and Introduction:

- in the sentence "Multifilm mass-transfer theory has been applied to food digestion in this research", consider rephrasing to clarify the objective and connect the idea to the hypothesis more effectively.

- Modify the phrase "Mass transfer coefficients (and resistances) have been analyzed in both in-vivo and in-vitro conditions" would enhance readability.

Grammar:

- Inconsistent use of hyphenation (e.g., in-vitro, two-film) needs to be standardized throughout the manuscript.

- Misplaced modifiers occasionally affect clarity, such as in "which are the intestine walls from animals.”

Figure Captions:

Descriptions in figure titles may be modified for clarity. "Figure 5 shows the intestine digestion model mimicking glucose generation and transport," write: "Figure 5 illustrates the intestine digestion model designed to replicate glucose generation and transport dynamics."

Authors are encouraged to make the following changes to the article:

Abstract

- Missing mention of specific applications or broader implications of the findings.

Introduction

- Some redundancies are present. It is recommended to modify “the comparison of in-vitro and in-vivo dynamics is mentioned multiple times.”

- The aim statement could be more precise. Replace “to understand the in-vitro carbohydrate digestion quantitatively” with “to quantitatively evaluate carbohydrate digestion in a novel in-vitro model.”

Materials and Methods

- Lack of a justification for certain experimental parameters, such as the chosen flow rates and enzyme concentrations.

- The description of the oral digestion model is too technical for readers unfamiliar with engineering aspects. Provide simpler explanations or analogies.

- Consider adding a brief subsection on statistical analysis, explaining how results were validated.

Results and Discussion

- The discussion of the Damköhler number (Da) is highly technical without sufficient context for readers unfamiliar with chemical kinetics.

- Repetition of results in both textual and tabular forms makes the section verbose. Glucose concentration data are described in detail alongside Table 5, which already summarizes the information.

- The significance of findings could be better linked to practical applications, such as improvements in dietary guidelines or food formulation.

Conclusion

- The conclusion is overly focused on technical aspects. Broader implications, such as potential applications in personalized nutrition or glycemic response prediction, should be discussed.

- Include a sentence on limitations and future research directions.

Round 2

Reviewer 2 Report

Comments and Suggestions for Authors

No comments